# LEARNING PRIORS FOR ADVERSARIAL AUTOENCODERS

**Hui-Po Wang, Wei-Jan Ko and Wen-Hsiao Peng**
National Chiao Tung University, Taiwan

## ABSTRACT

Recent studies show that the choice of the prior has a profound effect on the expressiveness of deep latent factor models. In this paper, we propose to learn the prior from data for adversarial autoencoders (AAEs). We introduce the notion of code generators to transform manually selected simple priors into ones that can better characterize the data distribution.

## 1 INTRODUCTION

Deep latent factor models, such as variational autoencoders (VAEs) and adversarial autoencoders (AAEs), involve specifying a prior distribution over latent variables and defining a deep generative network (i.e., the decoder) that maps latent variables to data space. Training such deep models usually requires learning a recognition network (i.e., the encoder) regularized by the prior. Traditionally, a simple prior, e.g. the standard normal distribution (Kingma & Welling, 2013), is used. However, some recent works (Hoffman & Johnson, 2016; Goyal et al., 2017; Tomczak & Welling, 2017) suggest that the choice of the prior may have a profound impact on the expressiveness of the model. Burda et al. (2015) indicate that the standard normal prior often results in overly regularized models. Hoffman & Johnson (2016) conjecture that multi-modal priors can achieve a higher variational lower bound on the data log-likelihood. Tomczak & Welling (2017) further confirm this conjecture by showing that their multi-modal prior consistently outperforms simple priors. Goyal et al. (2017) learn a tree-structured nonparametric Bayesian prior for capturing the hierarchy of semantics presented in the data. All these priors are learned under the VAE framework.

In this paper, we propose the notion of code generators for learning the prior from data for AAEs. The code generator, modeled by a neural network, is to transform a manually-specified simple prior into one that together with the decoder can better characterize the data distribution. To this end, we generalize the AAE framework in several significant ways: (a) we train a code generator to minimize an adversarial loss in data space; (b) we employ a learned similarity metric (Larsen et al., 2015) for training the autoencoder; and (c) we maximize the mutual information between part of the code generator input and the decoder output for supervised and unsupervised tasks using InfoGAN (Chen et al., 2016).

## 2 LEARNING THE PRIOR

To train the code generator, an objective function is needed to shape the distribution at its output. Normally, we wish to find a prior that, together with the decoder, would lead to a distribution that maximizes the data likelihood. We are however faced with two challenges. First, the output of the code generator could be any distribution, which makes the likelihood function and its variational lower bound intractable. Second, the decoder has to be learned simultaneously, which creates a moving target for the code generator.

To address the first challenge, we impose an adversarial loss on the decoder output when training the code generator. In symbols, this is to minimize

$$\mathcal{L}_{GAN}^I = \log(D_I(x)) + \log(1 - D_I(dec(z_c))), \tag{1}$$

where $z_c = CG(z)$ is the code generator output driven by a noise $z \sim p(z)$, $D_I$ is the discriminator in data space, and $dec(z_c)$ is the decoder output driven by $z_c$.

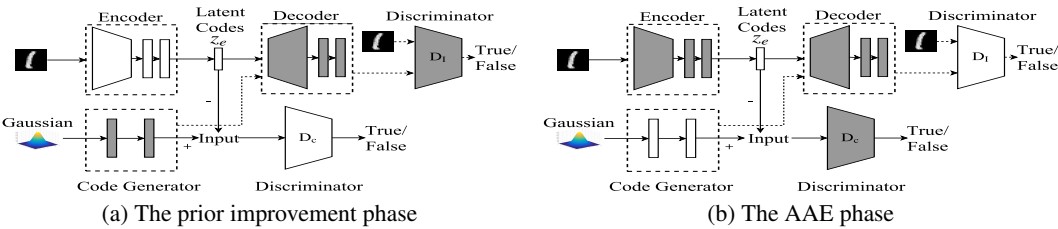

Figure 1: Training phases.

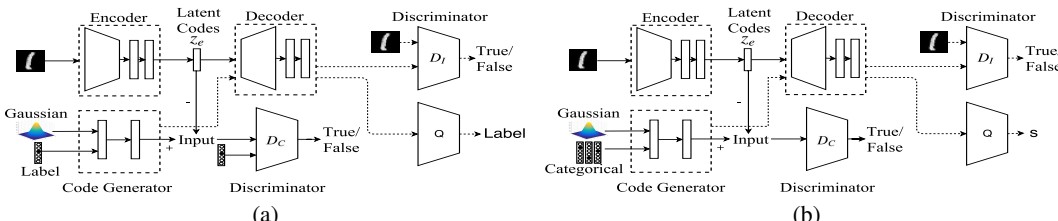

Figure 2: Providing conditions: (a) the supervised setting and (b) the unsupervised setting.

To address the second challenge, we propose to alternate training of the code generator and the decoder/encoder until convergence, as illustrated in Figure 1. In the prior improvement phase, we update the code generator based on minimizing Eq. (1) while fixing the encoder. In the regular AAE phase, we fix the code generator and update the autoencoder following the training procedure of AAE. Specifically, the encoder is regularized by the following adversarial loss in latent code space:

$$\mathcal{L}_{GAN}^C = \log(D_C(z_c)) + \log(1 - D_C(enc(x))),  \qquad (2)$$

where $enc(x)$ is the encoder output driven by $x$ and $D_C$ is the discriminator in latent space. Because the decoder will be updated in both phases, the convergence of the decoder relies on consistent training objectives during the alternation of training phases. The widely used pixel-wise squared error criterion in the AAE phase tends to produce blurry decoded images, which conflicts with the adversarial objective in the prior improvement phase. We thus adopt the learned similarity metric (Larsen et al., 2015) to compute the reconstruction errors in feature domain so that the decoder is driven consistently in both phases towards producing realistic images.

**Providing conditions**. In the following experiments, we showcase the ability of the proposed method to generate images conditionally on a control variable $s$ input to the code generator, as illustrated in Figure 2. With the supervised setting, the $s$ denotes the label associated with the input image, whereas with the unsupervised setting, it is governed by a categorical distribution. To have the code generator pick up the information carried by the variable $s$ when generating the latent code, we introduce the variational learning technique in InfoGAN (Chen et al., 2016) to maximize the mutual information $I(s; dec(z_c))$ between the variable $s$ and the generated image $dec(z_c)$.

## 3 EXPERIMENTS

This section demonstrates the superiority of the proposed model (AAE + learned similarity metric + learned prior) over a baseline model (AAE + learned similarity metric + fixed prior) in terms of learning disentangled representations on MNIST and CIFAR-10. Both models implement the same autoencoder with the ResNet and have the same latent space dimension (64-D), in order to understand the sole effect of the learned prior. The fixed prior regularizes the encoder output without being transformed by the code generator.

Figure 3 compares sample images generated by the decoder with the learned (proposed) and fixed (baseline) priors, respectively. Both models are trained under the supervised and unsupervised settings (cf. Providing conditions). On MNIST, both models work well in separating the label information from the remaining (style) information. On CIFAR-10, ours generates semantically more discernible images that match the labels in the supervised setting, whereas in the unsupervised setting, both exhibit a tendency to cluster images with similar colors. Moreover, the learned prior

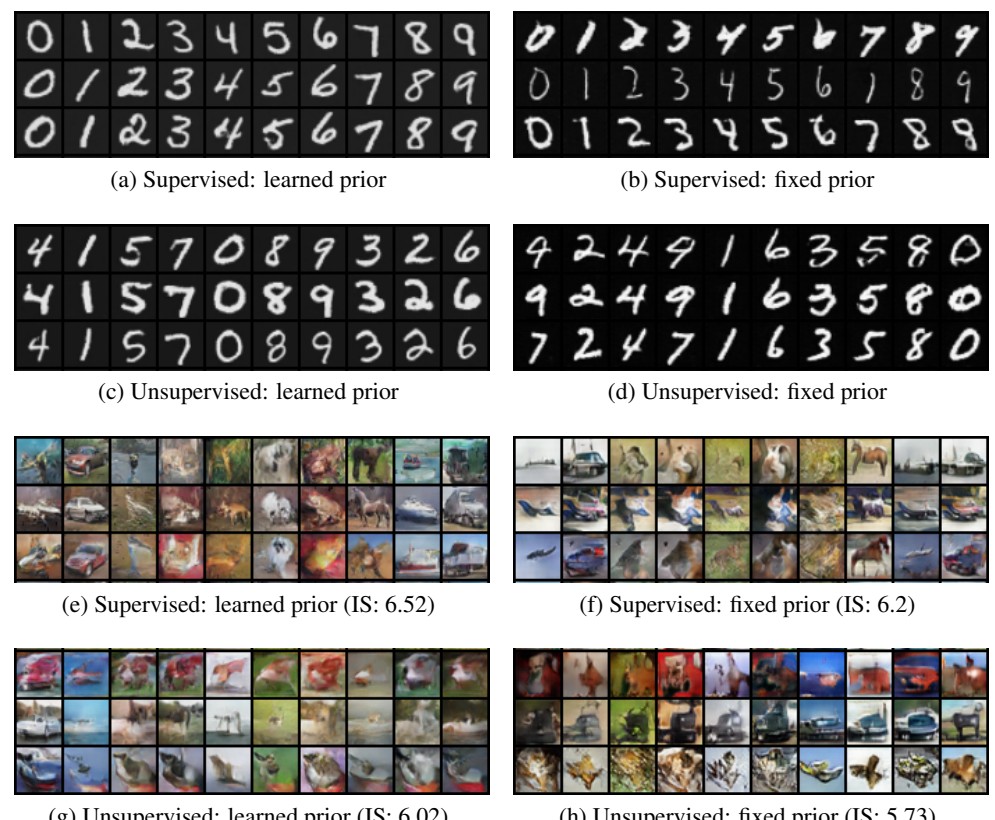

(a) Supervised: learned prior

(b) Supervised: fixed prior

(c) Unsupervised: learned prior

(d) Unsupervised: fixed prior

(e) Supervised: learned prior (IS: 6.52)

(f) Supervised: fixed prior (IS: 6.2)

(g) Unsupervised: learned prior (IS: 6.02)

(h) Unsupervised: fixed prior (IS: 5.73)

Figure 3: Sample images produced by the proposed and baseline models trained supervisedly and unsupervisedly on MNIST and CIFAR-10. Each column of images have the same label/class information but varied Gaussian noise. Inception scores (IS) are presented for CIFAR-10 images.

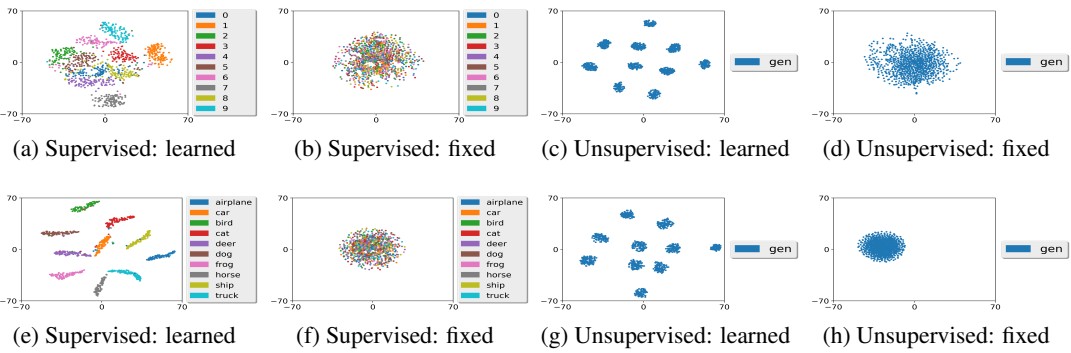

(a) Supervised: learned    (b) Supervised: fixed    (c) Unsupervised: learned    (d) Unsupervised: fixed

(e) Supervised: learned    (f) Supervised: fixed    (g) Unsupervised: learned    (h) Unsupervised: fixed

Figure 4: Latent code space visualization on MNIST (upper) and CIFAR-10 (lower).

consistently outperforms the fixed prior in terms of inception scores. Figure 4 further visualizes their latent code space with T-SNE (Maaten & Hinton, 2008). It is seen that the learned prior forms obvious clusters according to the label/class in both settings, presenting its regularization effect on the encoder output. On the contrary, the fixed prior, although clustering in high-dimensional space along the label/class-defining dimensions, shows a uni-modal distribution in 2-D space due to the limitations of T-SNE. These confirm the benefits of the learned prior in generating better quality images and learning better disentangled representations than the fixed prior. For complete experimental results, please visit our website[1].

---

[1]https://github.com/a514514772/Learning-Priors-for-Adversarial-Autoencoders

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
