# OpenReview forum: "Learning Priors for Adversarial Autoencoders"
_ICLR.cc/2018/Workshop — Reject_

### Official Review · AnonReviewer2 · 2018-03-08
**Review: Sensible but incremental proposal**

**Rating:** 4
**Confidence:** 4

**Review:**


Summary

This paper proposes to parameterize the prior of the adversarial autoencoder (AAE).  The usual standard Gaussian is transformed by one or more neural network layers, resulting in a prior that is better dispersed and has multiple modes (according to t-SNE visualizations).

Evaluation

Pros:  Parametrizing the prior of deep generative models is a sensible thing to do, and the approach seems to result in improvements to the generated samples and to the latent space (as seen in t-SNE visualizations) in comparison to a fixed Gaussian prior.

Cons:  Parametrizing the prior is an incremental suggestion and not particularly novel.  The authors note this themselves, citing the work on VampPriors [2].  Yet, the authors fail to cite the work on lossy autoencoders (LAEs) [1], which also parametrizes the prior.  The only difference is that LAEs use an autoregressive flow because they require the prior to remain a tractable density whereas this work can employ a non-invertible map because of its use of an adversarial loss.  Regarding experiments, I think a better baseline would be to compare against a fixed mixture density prior.  Doing so would test if the additional parameters (which need to be optimized) in the prior’s transformation are necessary.

Conclusion

While I find no serious flaws in this work, I can’t recommend its acceptance due to a lack of novelty.


1.  X. Chen, D. Kingma, T. Salimans, Y. Duan, P. Dhariwal, J. Shulman, I. Sutskever, and P. Abbeel.  “Variational Lossy Autoencoder.”  ICLR 2017.

2.  J. Tomczak and M. Welling.  “VAE with a VampPrior.”  AIStats 2018.

---

### Official Review · AnonReviewer3 · 2018-03-10
**LEARNING PRIORS FOR ADVERSARIAL AUTOENCODERS**

**Rating:** 4
**Confidence:** 4

**Review:**

The authors consider the problem of learning the prior distribution in adversarial autoencoder. To do so, the authors, specify a neural network code generator that transform a simple prior distribution into a more flexible representation.

The authors do not describe how to incorporate the control variable, s, in AAE. That being said, it is not entirely clear whether the proposed model outperforms AAE because of the code generator or the added information from the control variable.

Please clarify in Figure 1 that grey and white boxes indicate fixed and updated modules, respectively. The Figures could also be simplified to improve readability.

---

### Official Review · AnonReviewer1 · 2018-03-11
**Interesting approach to improving adversarial autoencoders**

**Rating:** 6
**Confidence:** 3

**Review:**

The paper proposes improving adversarial autoencoders by learning a transformation of the prior distribution to be one that produces images that can fool a standard GAN discriminator. The results are interesting though preliminary.

The results would be more convincing if the paper showed that interpolating between different samples from the prior produced images that change smoothly since otherwise the transformation on the prior distribution could simply map everything to the latent codes in the training set.

---

### Decision · Program_Chairs · 2018-03-20
**ICLR 2018 Workshop Acceptance Decision**

**Decision:**

Reject

**Comment:**

Based on the reviews, this paper has not been accepted for presentation at the ICLR workshop. However, the conversation and updates can continue to appear here on OpenReview.